# Identification of *Prototheca* from the Cerebrospinal Fluid of a Cat with Neurological Signs

**DOI:** 10.3390/vetsci10120681

**Published:** 2023-11-30

**Authors:** Gianvito Lanave, Francesco Pellegrini, Giuseppe Palermo, Eric Zini, Edy Mercuriali, Paolo Zagarella, Krisztián Bányai, Michele Camero, Vito Martella

**Affiliations:** 1Department of Veterinary Medicine, University of Bari Aldo Moro, Valenzano, 70010 Bari, Italy; gianvito.lanave@uniba.it (G.L.); francesco.pellegrini@uniba.it (F.P.); vito.martella@uniba.it (V.M.); 2Veterinary Orthopaedic Traumatologic Centre of Arenzano, Arenzano, 16011 Genova, Italy; giuseppepalermo.medvet@gmail.com (G.P.); edy.mercuriali@ctovet.com (E.M.); paolo.zagarella@ctovet.com (P.Z.); 3Veterinary Institute of Novara, Granozzo con Monticello, 20060 Novara, Italy; eric.zini@unipd.it; 4Department of Animal Medicine, Production and Health, University of Padova, 35020 Legnaro, Italy; 5Clinic for Small Animal Internal Medicine, Vetsuisse Faculty, University of Zurich, CH-8057 Zurich, Switzerland; 6Veterinary Medical Research Institute, Hungarian Academy of Sciences, H-1143 Budapest, Hungary; banyai.krisztian@vmri.hun-ren.hu; 7Department of Pharmacology and Toxicology, University of Veterinary Medicine, H-1078 Budapest, Hungary

**Keywords:** feline, prototheca, neurological symptoms, infection, alga, diagnosis

## Abstract

**Simple Summary:**

The genus *Prototheca* encompasses unicellular algae that are achlorophyllous and widespread in the environment. The genus is now included in the family *Chlorellaceae,* belonging to the order *Chlorellales,* which is included in the class, *Trebouxiophyceae*. Prototheca have repeatedly been reported to infect vertebrates. Cattle, dogs, and cats are the unique domestic animals in which *Prototheca* spp. have been reported, despite sporadic detection in goats, horses, and non-domesticated animals. *Prototheca* spp. have been reported to colonize different districts of the human body. Cats with protothecosis usually display a cutaneous disease, whereas dogs may develop both cutaneous and systemic forms. In this report, we identified molecularly *Prototheca* spp. in a cat with neurological signs. The animal presented a suspected diagnosis of multifocal lymphoma, and eventual immunological disorders/suppression likely triggered systemic diffusion of the achlorophyllic alga. Despite protothecosis not being regarded as a zoonosis, algal infections of animals should be recognized as indicators or sentinels of environmental risks for humans.

**Abstract:**

Prototheca infections are rare in cats, and they are usually associated with cutaneous or subcutaneous infections by *P. wickerhamii*, with no evidence of neurological signs or systemic disease. In this study, we report the identification of prototheca in the cerebrospinal fluid (CSF) of a cat with neurological symptoms. Fourteen CSF samples were gathered from cats presented with neurological disease between 2012 and 2014. The inclusion criteria for the samples were an increase in CSF protein and cell number (pleocytosis), suggestive of an infectious inflammatory status of the central nervous system (CNS). Nine samples fulfilled the inclusion criteria (inflammatory samples), while five samples, used as control, did not (non-inflammatory samples). All the samples were screened molecularly for different pathogens associated with CNS disease in cats, including prototheca. Out of 14 CSF samples, only one inflammatory sample tested positive for prototheca. Upon sequence and phylogenetic analysis of the amplicon, the strain was characterized as *P. bovis*. This report is the first documented evidence of prototheca in the cerebrospinal fluid of a cat with neurological signs. Prototheca should be considered in the diagnostics procedures on the CNS of cats presented with infectious diseases.

## 1. Introduction

The *Prototheca* spp. consist of microscopical and unicellular organisms that are obligatory heterotrophs because they lack chloroplasts capable of photosynthesis [1,2,3]. Despite their yeast-like morphology, based on genetic features, *Prototheca* spp. have been classified as algae and included in the *Prototheca* genus closely related to *Chlorella* genus in the family, *Chlorellaceae* [4]. *Prototheca* spp. are ubiquitous, may also colonize animal and human gastrointestinal tracts, and have been occasionally reported in the skin and nail beds of asymptomatic human patients [1,3,5]. *Prototheca* spp. Are also able to infect animals, but their specific pathogenic mechanisms of infection are yet to be elucidated. Several *Prototheca* spp., i.e., *P. cutis*, *P. miyajii*, *P. ciferrii*, *P. wickerhamii*, *P. bovis*, and *P. blaschkeae*, are able to infect both humans and animals [6,7,8].

Protothecosis is a rare and occasional disease reported in humans and domesticated as well as wild animals. Human and canine infections have been described worldwide [9]. Mucosal contact, ingestion, or traumatic introduction from contaminated fonts are regarded as the most common sources of transmission of *Prototheca* spp. The algae penetrate the body via the respiratory or gastrointestinal tract and may then diffuse via ocular, cerebral, and renal routes [10,11]. Over 95% of infections in human patients are due to *P. wickerhamii*, with a small number of cases by *P. bovis*, *P. miyajii*, *P. blaschkeae*, *P. ciferri*, or *P. cutis* [12,13,14]. In dogs, most prototheca infections are caused by *P. zopfii*, with a few cases due to *P. wickerhamii* [11,15].

Feline protothecosis is quite infrequent, either due to natural resistance to infection or circumvention of environmental niches where algae commonly establish. The exiguous recorded cases have all been reported in clinically healthy adult cats with solid, non-ulcerated, cutaneous or subcutaneous masses located on the forehead, distal limbs, base of the tail, nose, or pinnae [16,17,18,19], and when the isolates have been speciated, they have all been characterized as *P. wickerhamii* [20]. Nasal localization of prototheca has also been reported in cats [15,21]. The lack of regional lymphadenomegaly and clinical signs associated with systemic infection/disease suggests that in cats, prototheca infection tends to be localized [10], although a unique cat displayed new distant nodules several months after excisional biopsy of an original solitary lesion [19]. Accordingly, unlike dogs, there is no evidence in the literature for neurological signs or systemic symptoms associated with prototheca infection in cats [10].

## 2. Materials and Methods

### 2.1. Collection of Samples

Fourteen cerebrospinal fluid (CSF) samples were gathered from cats with neurological disease at the veterinary clinics of Novara and Arma di Taggia, Imperia, Italy, between 2012 and 2014. The inclusion criteria for the samples were raised CSF protein and an increase in the CSF cell number (pleocytosis), parameters suggestive of an infectious inflammatory status of the central nervous system (CNS). Pleocytosis in a CFS sample (lumbar punctate) was categorized as positive with a protein fraction > 30 mg/dL or number of cells > 3 cells/µL, with a predominance of mononuclear cells. Nine samples fulfilled the inclusion criteria (inflammatory samples), while 5 samples did not and they were used as control. 

### 2.2. Nucleic Acid Extraction

The nucleic acids were subjected to extraction from CSF samples employing the IndiSpin^®^ Pathogen Kit (Indical Bioscience GmbH, Leipzig, Germany), following the manufacturer’s instructions. Nucleic acid templates were stored at –70 °C until use.

### 2.3. Screening for Prototheca spp.

Nucleic acid extracts were subjected to a PCR assay specific for the 18SrDNA of prototheca, using the forward primer Proto 18S-4F (5′-GACATGGCGAGGATTGACAGA-3′) and the reverse primer Proto 18S-4R-1 (5′-ATCACAGACCTGTTATC-3′) [22,23], which amplify a PCR product of approximately 250 bp (Table 1). 

Amplification was conducted using the Accuprime PCR Kit (Invitrogen^TM^ Thermo Fisher Scientific, Shanghai, China) and *P. blaschkeae* as a positive control. The bands were subjected to excision and purification by a QiaQuick Gel Extraction Kit (Qiagen GmbH, Hilden, Germany), and the sequence was determined. Sequencing was performed at Eurofins Genomics (Vimodrone, Milano) laboratories. As an internal control, primers targeting the 28S rRNA gene of the feline genome were used [33]. 

### 2.4. Quantitative Real Time PCR (qPCR), Specifically for P. bovis

A qPCR specific for *P. bovis* was performed on samples testing positive for *Prototheca* spp. Ten μL of sample DNA was combined with the 15-μL reaction master mix (IQ Supermix; Bio-Rad Laboratories SRL, Segrate, Italy), comprising 0.6 μmol/L of each primer and 0.2 μmol/L of the probe (Table 1). Thermal cycling was performed according to a previously described study [24].

### 2.5. Screening for Other Pathogens

Nucleic acid extracts were also screened for other feline pathogens, including feline infectious peritonitis, feline leukemia virus (FeLV), feline immunodeficiency virus (FIV), feline panleukopenia virus, rickettsia, neospora, toxoplasma, mycobacterium, and bacterial 16S rDNA (Table 1).

### 2.6. Sequence and Phylogenetic Analyses

The online tool BLAST (https://blast.ncbi.nlm.nih.gov/Blast.cgi, accessed on 15 September 2023) was used to find the highest nt identity in the NBCI database. Sequence editing was performed by the software package Geneious Prime v. 2021.2 (Biomatters Ltd., Auckland, New Zealand). Sequence alignments were performed by the MAFFT [34] plugin implemented in Geneious Prime. The best-fitting substitution model settings for the phylogeny were explored by the tool “Find the best protein DNA/Protein Models” of the MEGA X v. 10.0.5 software [35]. The evolutionary history was deduced by using the maximum-likelihood method, the Kimura 2-parameter model, a discrete gamma distribution to model evolutionary rate differences among sites (6 categories), and supplying statistical support with 1000 replicates. Bayesian inference and neighbor-joining phylogenetic analyses were also performed to explore the phylogeny of *Prototheca* spp.

## 3. Results

Fourteen CSF samples collected in this study were subjected to molecular screening for *Prototheca* spp. and a panel of feline pathogens. Out of 14 CSF samples, 1 sample (#628/14) tested positive for *Prototheca* spp. by PCR, and the sequence was determined. By BlastN analysis performed on a 250 bp sequence of 18SrDNA of *Prototheca* spp., strain #628/14 shared the highest nucleotide (nt) identity (100%) with *P. zopfii var. hydrocarbonea* strain UP-PT-P1 (EU439263). All the samples tested negative for feline infectious peritonitis, feline leukemia virus, feline immunodeficiency virus, feline panleukopenia virus, rickettsia, neospora, toxoplasma, mycobacterium, and bacterial 16S rDNA. 

Partial 18SrDNA sequence (250 nt)-based phylogenetic analysis was performed using the sequence of *Prototheca* spp. generated in the study and the cognate sequences of the closest relatives retrieved from the NCBI database. Different phylogenetic approaches were explored for *Prototheca* spp., and similar topologies with slight differences in bootstrap values at the nodes of the tree were noticed. Accordingly, the maximum-likelihood (ML) tree was used. Upon ML analysis, strain ITA/2014/628 segregated with strains belonging to the *P. bovis* clade (Figure 1). Upon qPCR specific for *P. bovis*, sample #628/14 yielded 27 Ct. 

The animal that tested positive for *Prototheca* spp. was a 9-year-old male domestic European cat, presented at the veterinary clinic with a 24 h history of seizures, incoordination, circling, and disorientation. Clinical pathological evaluation included a complete blood count and clinical chemistry panel. Blood analysis showed a marked increase in creatine kinase (9827 U/L, reference interval [ref.]: 91–326 U/L), alanine aminotransferase (517 U/L, ref.: 22–45 U/L), and aspartate aminotransferase (98 U/L, ref.: 21–41 U/L). The other parameters were not altered. Complete blood count parameters were within the reference interval. In addition, at the time of clinical examination, the animal tested negative for FIV and FeLV, using a quick test (SNAP FIV/FeLV Combo Test—IDEXX Laboratories). Abdominal ultrasound examination revealed multiple spleen and liver nodules. Fine needle biopsy specimens taken from the spleen and liver nodules revealed many lymphoid cells, and a suspect of lymphoma was included in the differential diagnosis. 

Within 24 h, the clinical condition of the animal worsened, and the owner opted for the gentle suppression of the animal. Extreme care was employed to guarantee that death had happened prior to discarding the animal remains [36]. Before euthanasia, a CSF sample (#628/14) was collected with the permission of the owner, exclusively for research purposes. 

Examination of the CSF displayed a distinct increase in total protein (2432 mg/L, range <300 mg/L) and cytological features consistent with marked mixed-type pleocytosis (2448 cells/µL, range <3 cells/µL), composed mainly of small and medium-sized lymphocytes with no red blood cells. The owner did not give permission for further investigations (i.e., necropsy) in the animal.

## 4. Discussion

The taxonomic status of prototheca has been revised in the last few decades. Originally, seven species have been designated for the genus, namely *P. stagnora*, *P. ulmea*, *P. wickerhamii*, *P. blaschkeae*, *P. zopfii*, *P. cutis*, and *P. miyajii* [22,37,38]. *P. blaschkeae*, originally considered as biotype 3 of *P. zopfii*, has been re-classified as a new species, while *P. zopfii* biotypes 1 and 2 have been genetically correlated to distinct genotypes, I and II, respectively [22]. *P. moriformis* has been subsequently recognized as a distinct species [22,39]. However, in the late 2010s, a novel classification of the genus *Prototheca* was proposed [7], adding six new species (*P. cookei* sp. nov., *P. cerasi* sp. nov., *P. pringsheimii* sp. nov., *P. xanthoriae* sp. nov., *P. tumulicula*, and *P. stagnora*). Two main lineages were included in the genus: the triad composed of *P. ciferrii* sp. nov. (previously regarded as *P. zopfii* genotype I), *P. zopfii* (genetically distinct from *P. ciferrii* sp. nov.), *P. bovis* sp. nov. (previously named *P. zopfii* genotype II), and *P. wickerhamii* cluster. However, three novel species (*P. cookei* sp. nov., *P. cerasi* sp. nov., and *P. pringsheimii* sp. nov.) were related to the triad despite some important differences. Moreover, *P. wickerhamii*, *P. cutis*, *P. miyajii*, and the newly proposed species, *P. xanthoriae* sp. nov., have been regarded as closely related to the genera Chlorella and Auxenochlorella.

In this study, we identified prototheca in the CSF of a cat with neurological signs. Upon sequence and phylogenetic analysis, the feline strain 628/14 was characterized as *P. bovis*. *P. bovis* is responsible for most (i.e., 75 to 90%) of prototheca infections in dogs [10], of which skin tissue, digestive, nervous and ophthalmic systems [11,40,41,42,43,44,45,46], or localized changes in the nervous system or enteric tract have been reported [46,47]. Feline protothecosis seems to be very rare, and *P. wickeramii* has been mainly reported in cutaneous disease in cats [16,18,19,20,48], with only sporadic detection of *P. bovis* in a cat with nasal dermatitis [21]. 

In humans, protothecosis has been described in three main clinical forms, namely skin lesions, olecranon bursitis, and disseminated/systemic infections [5,49]. The cutaneous form seems more common form of protothecosis [50]. Additionally, an immunosuppressive factor has been identified in about half of human infections. *P. bovis* seems to be the predominant prototheca species in systemic infections, while *P. wickerhamii* is mostly involved in skin infections. Workers in animal productions or with food products of animal origin may be at a higher risk of exposure to protothecal infections [5]. Overall, protothecosis is considered a rare human disease with an increased incidence in patients with immunosuppression [50]. Epidemiological investigations based on a One Health approach, including humans, animals, and the environment, would be necessary to better understand the ecology of prototheca [51].

Algal invasion of the CNS is usually secondary to primary replication and lesions in other districts. The cat positive for prototheca in this study did not have a history of cutaneous disease, which is the common clinical picture associated with prototheca in cats. However, the animal had a suspected diagnosis of multifocal lymphoma, and therefore it was likely affected by immunological disorders/suppression, facilitating the infection and systemic spread of the algal pathogen. 

Our study was affected by several limits. For instance, we failed to isolate the algal strain in cultural media. This was likely attributed to the fact that the sample was stored at −80° C for a long period before testing and isolation attempts. Sabouraud dextrose agar culture would have provided a firmer diagnosis [52], although molecular diagnostics have become standard in recent years [53,54]. Additionally, we could not further analyze the tissues/organs of the animal since the owners did not give permission for necropsy, and the retrospective nature of our investigation did not allow us to carry out other diagnostic investigations. CSF analysis is often inconclusive and should be used in combination with other approaches in the diagnostic pipeline for neurological diseases. The results obtained in this report should be further confirmed in other independent studies in order to understand the role of prototheca in neurological diseases of cats. 

Prototheca was identified in this study through PCR targeting the 18SrDNA. When performing molecular screenings, we made several efforts to avoid laboratory contamination with DNA, enacting rigorous laboratory practices. The development of molecular assays for identification of prototheca allows us to expand the literature and records of algal infection in cats and dogs. Misdiagnosis with viral or bacterial pathogens based on the clinical/biochemical picture can be reduced by implementing diagnostic for these pathogens into microbiological laboratories. 

## 5. Conclusions

Our findings indicate that prototheca should be considered in the diagnostic algorithms for infectious CNS diseases in cats. Moreover, while protothecosis is not considered a zoonosis, algal infections in animals should be considered as indicators or sentinels of environmental risks for humans.

## Figures and Tables

**Figure 1 vetsci-10-00681-f001:**
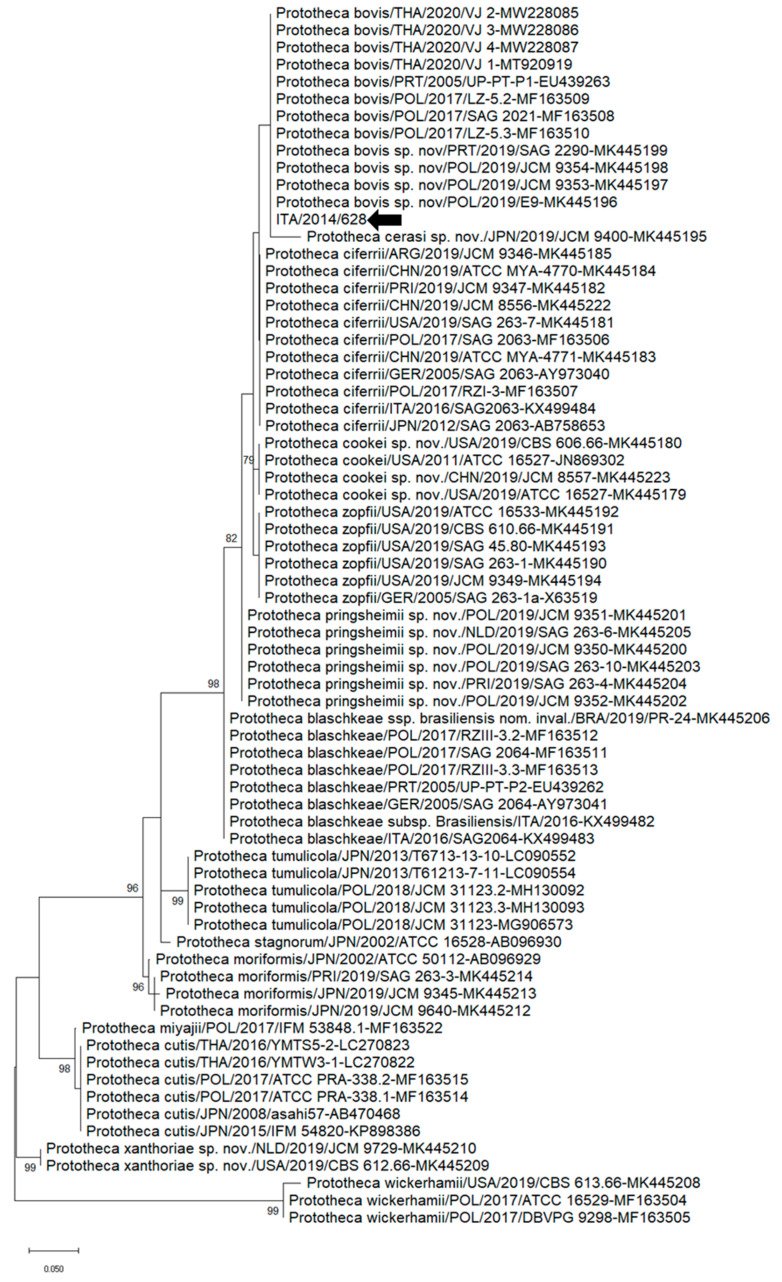
Partial 18SrDNA sequence (250 nt)-based phylogenetic tree of *Prototheca* spp. strains reported in this study and reference strains retrieved from the NCBI database. The Maximum Likelihood method and Kimura model (two parameters), with a discrete gamma distribution, were applied for the phylogenic analysis. One thousand bootstrap replicates were employed to assess the robustness of individual nodes on the phylogenetic tree. Bootstrap values higher than 75% were displayed. Black arrows denote strains reported in this study. Numbers of nucleotide substitutions are connoted by the scale bar.

**Table 1 vetsci-10-00681-t001:** Detailed list of protocols (pathogens, assays, primers, probes, and references) used for the molecular screening of samples included in the study.

Pathogen	Assay	Primers and Probes	Oligonucleotide Sequence	Reference(s)
*Prototheca* spp.	PCR	Proto 18S-4F Proto 18S-4R-1	5′-GACATGGCGAGGATTGACAGA-3′ 5′-ATCACAGACCTGTTATC-3′	[22] [23]
*Prototheca bovis*	qPCR	PZg2F SPZg2 PZg2R	5′-GACGATGATCCTAGTTATGGTGTAC-3′	[24]
			5′Fam-TGGTAGAAGACAAATAATGTACCAAAACCA-BHQ13′	
			5′-TATAAAAGCAAGTCCAGTTACAGCAC-3′	
feline infectious peritonitis	qPCR	FCoV1128f FCoV1200p FCoV1229r	5′-GATTTGATTTGGCAATGCTAGATTT-3′ 5′Fam-TCCATTGTTGGCTCGTCATAGCGGA-Tamra3′ 5′-AACAATCACTAGATCCAGACGTTAGCT-3′	[25]
feline leukemia virus	PCR	118for 119rev	5′-TTACTCAAGTATGTTCCCATG-3′ 5′-CTGGGGAGCCTG GAGACTGCT-3′	[26]
feline immunodeficiency virus	PCR	158for 159rev	5′-GAGTAGATACWTGGTTRCAAG-3′ 5′-CATCCTAATTCTTGCATAGC-3′ 5′-CAAAATGTGGATGGTGGAAY-3′ 5′-ACCATTCCWATAGCAGTRGC-3′	[27]
	nPCR	160for 161rev		
feline panleukopenia virus	qPCR	FPV/CPV-For FPV-Pb CPV-Pb FPV/CPV-Rev	5′-ACAAGATAAAAGACGTGGTGTAACTCAA-3′ 5′Vic-ATGGGAAATACAGACTATAT-MGB3′ 5′Fam-ATGGGAAATACAAACTATAT-MGB3′ 5′-CAACCTCAGCTGGTCTCATAATAGT-3′	[28]
rickettsia	PCR	RSFG 877 RSFG1258	5′-GGGGGCCTGCTCACGGCGG-3′ 5′-ATTGCAAAAAGTACAGTGAACA-3′	[29]
neospora	qPCR	Neo For Neo Probe Neo Rev	5′-GCATCGGAGGACACTGCT-3′ 5′Fam-CTGACTCTGAACACCGGAGGCACG-Tamra3′ 5′-ATGTCGTAAATCGGAGTTGCTTC-3′	[30]
toxoplasma	qPCR	Tox For Tox Probe Tox Rev	5′-GTCCTATCGCAACGGAGTTCTT-3′ 5′Fam-CCAGACGTGGATTTCCGTTGGTTCC-Tamra3′ 5′-TTCGTCCGTCGTAATATCAGGC-3′	
mycobacterium	PCR	Myc For Myc Rev	5′-CATGCAAGTCGAACGGAAAG-3′ 5′-CGGTGCTTCTTCTCCACCTA-3′ 5′-TACTCGAGTGGCGAACGGGT-3′ 5′-CGGACCTTCGTCGATGGTGA-3′	[31]
	nPCR	Myc NFor Myc NRev		
bacterial 16S rDNA	PCR	B-V5 A-V6	5′-ATTAGATACCCYGGTAGTCC-3′ 5′-ACGAGCTGACGACARCCATG-3′	[32]
Internal control	PCR	feline 28S rDNA Fw	5’-AGCAGGAGG TGTTGGAAGAG-3′	[33]
		feline 28S rDNA Rv	5′-AGGGAGAGCCTAAATCAAAGG-3′	

## Data Availability

Data are contained within the article.

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
