# Peer review of "Identification of Prototheca from the Cerebrospinal Fluid of a Cat with Neurological Signs"

_vetsci, 2023, doi:10.3390/vetsci10120681_

Round 1
Reviewer 1 Report
Comments and Suggestions for Authors
This article describes the finding of the alga Prototheca bovis in a cerebrospinal fluid (CSF) sample obtained from a cat with neurological symptoms from a collection of 14 frozen feline CSF samples obtained years ago. The sample had been frozen for several years before analysis, and a necropsy of the cat could not be performed.
One of the strengths of this article is that a correct analysis and methodology were applied to the samples, including the most common diseases that produce neurological symptoms in cats.
However, it should be noted that these samples only allow for the exclusion of these diseases in the CSF, not in the animal under study. It is worth mentioning that CSF analysis often yields unremarkable or normal results, and therefore should not be used in isolation as a diagnostic technique for neurological diseases.
Although it has been described that Prototheca spp. can cause neurological lesions in dogs, there is no scientific evidence in the literature regarding cats. Therefore, we must be very strict when concluding that the presence of this alga in the CSF is the cause of the lesions and symptoms in the study patient. Support from other diagnostic techniques such as advanced imaging, cultures, and pathological studies are essential for reaching definitive diagnoses, as can be found in this publication.
DOI: 10.1111/j.1939-165X.2011.00395.x
There is no alignment between the finding of the alga in the sample and the symptoms, as it cannot be proven that the alga was responsible for the lesions in the nervous tissue. The mere presence of the alga in the sample does not justify the conclusion, as it could be due to other reasons, such as sample contamination, processing contamination (although this is unlikely, it is possible), contamination of the tube in which the sample is stored (for long period), or simply an incidental finding unrelated to the symptoms (in fact, the authors themselves suggest that the animal could be immunosuppressed and that the cat presented lesions consistent with lymphoma, which could explain the symptoms -i.e. DOI: 10.3390/ani13050862, and many others-, but a definitive diagnosis was not reached (no necropsy was performed, and there were no descriptive lesions indicating the effect of the alga on the affected nervous tissue).
This article does not require a specific expanded statistical analysis.
A higher level of evidence is required to reach the conclusion of this study.
Author Response
Dear Referee,
herein you can find a point-by-point response to the comments and suggestions.
R1.1 This article describes the finding of the alga Prototheca bovis in a cerebrospinal fluid (CSF) sample obtained from a cat with neurological symptoms from a collection of 14 frozen feline CSF samples obtained years ago. The sample had been frozen for several years before analysis, and a necropsy of the cat could not be performed. One of the strengths of this article is that a correct analysis and methodology were applied to the samples, including the most common diseases that produce neurological symptoms in cats. However, it should be noted that these samples only allow for the exclusion of these diseases in the CSF, not in the animal under study. It is worth mentioning that CSF analysis often yields unremarkable or normal results, and therefore should not be used in isolation as a diagnostic technique for neurological diseases.
Reply to R1.1 We thank the referee for the appreciation of the analysis and methodology applied to samples. We agree with the referee’s comment. We o added in the text a sentence to remark that “CSF analysis often is not conclusive, and therefore should be used in combination with other approaches in the diagnostic pipeline of neurological diseases (page 7, revised manuscript)”.
R1.2 Although it has been described that Prototheca spp. can cause neurological lesions in dogs, there is no scientific evidence in the literature regarding cats. Therefore, we must be very strict when concluding that the presence of this alga in the CSF is the cause of the lesions and symptoms in the study patient. Support from other diagnostic techniques such as advanced imaging, cultures, and pathological studies are essential for reaching definitive diagnoses, as can be found in this publication.
DOI: 10.1111/j.1939-165X.2011.00395.x
Reply to R1.2: we agree with the referee’s comment, although the nature of these observations is expected to be rare/sporadical and therefore the literature is scanty. We disclosed in the discussion that our report should be further confirmed in other independent studies in order to understand the role of prototheca in neurological diseases of cats (page 7, revised manuscript).
R1.3 There is no alignment between the finding of the alga in the sample and the symptoms, as it cannot be proven that the alga was responsible for the lesions in the nervous tissue. The mere presence of the alga in the sample does not justify the conclusion, as it could be due to other reasons, such as sample contamination, processing contamination (although this is unlikely, it is possible), contamination of the tube in which the sample is stored (for long period), or simply an incidental finding unrelated to the symptoms (in fact, the authors themselves suggest that the animal could be immunosuppressed and that the cat presented lesions consistent with lymphoma, which could explain the symptoms -i.e. DOI: 10.3390/ani13050862, and many others-, but a definitive diagnosis was not reached (no necropsy was performed, and there were no descriptive lesions indicating the effect of the alga on the affected nervous tissue). This article does not require a specific expanded statistical analysis. A higher level of evidence is required to reach the conclusion of this study.
Reply to R1.3: we agree with the referee that there is not a clear demonstration/correlation between the PCR results and the disease. We added in the text a sentence to stress this point (see reply to point R1.2). Yet we would like to mention that even the other samples were in the same storage conditions as the prototheca-positive sample. Also, in our defense, we would like to point out that CSF s is a common target for prototheca or for other pathogens for diagnostic PCR, i.e. the research of prototheca in human and animal samples (doi: 10.1111/j.1939-165X.2011.00395.x.; doi: 10.2147/IDR.S320795.). Thefore, our approach mirrors what reported in the literature.
Reviewer 2 Report
Comments and Suggestions for Authors
It is requested from the authors to provide evidence (PCR gel images and qPCR read outs) of the amplifications performed using test and control primers. Please enlist the internal and/or positive control primers used in Table 1. The author is also requested to provide some estimation or quantification of presence of Prototheca spp. over control.
It is strongly advised to either remove 2.6 Attempt of isolation, or support this method with the figures of microscopic colonies of Prototheca spp. The failure of this attempt has been mentioned in the discussion section. The authors are advised to discuss the potential reasons of failure in the attempts and further improvements that can be performed in such cases in the future.
The authors have mentioned that a fine needle biopsy of the european cat's spleen and liver resulted in the presence of nodules and that has suspected diagnosis of lymphoma. Is there any correlation of lymphoma with Prototheca spp infection? Can the neurological symptoms be associated with pre-exsisting conditions rather than Prototheca spp infection? Are there any known neurological symptoms reported in dogs and suspected to be in cats?
Comments on the Quality of English Language
Please recheck the manuscript for minor typographical and grammatical errors. Example - line 131 - 0.3 mL/kg.
Author Response
Dear Referee,
herein you can find a point-by-point response to the comments and suggestions.
R2.1 It is requested from the authors to provide evidence (PCR gel images and qPCR read outs) of the amplifications performed using test and control primers. Please enlist the internal and/or positive control primers used in Table 1. The author is also requested to provide some estimation or quantification of presence of Prototheca spp. over control.
Reply to R2.1. We provided the requested information in Table 1 and in the text (pag 3, revised manuscript). As internal control in feline samples, we use primers targeting the feline 28S rRNA gene. The primers (feline 28S rDNA Fw 50-AGC AGG AGG TGT TGG AAG AG-30 and feline 28S rDNA Rv 50-AGG GAG AGC CTA AAT CAA AGG-30) were previously described by Helps and collaborators (2003).
As we did not have a quantitative PCR (i.e., Taqman) for prototheca, we adopted a protocol available in the literature (Bacova et al., 2021). With this protocol, the positive control was recognized at 24 Ct whilst the feline sample at 27 Ct. We included this information in the manuscript (pag 4, revised manuscript).
R2.2 It is strongly advised to either remove 2.6 Attempt of isolation, or support this method with the figures of microscopic colonies of Prototheca spp. The failure of this attempt has been mentioned in the discussion section. The authors are advised to discuss the potential reasons of failure in the attempts and further improvements that can be performed in such cases in the future.
Reply to R2.2. We removed the section from Materials and Methods. We left a comment in the discussion: “For instance, we failed to isolate the algal strain in cultural media. This was likely attributed to the fact that the sample was stored at -80° C for a long period before being tested and used for the isolation attempts. Sabouraud dextrose agar culture would have provided a firmer diagnosis [49], although molecular diagnostic has become a standard in recent years [50,51]” (page 7, revised manuscript).
R2.3 The authors have mentioned that a fine needle biopsy of the european cat's spleen and liver resulted in the presence of nodules and that has suspected diagnosis of lymphoma. Is there any correlation of lymphoma with Prototheca spp infection? Can the neurological symptoms be associated with pre-exissting conditions rather than Prototheca spp infection? Are there any known neurological symptoms reported in dogs and suspected to be in cats?
Reply to R2.3: the referee’s comments/questions are surely interesting and it would be interesting to explore/investigate further these topics. However, considering the occasional/sporadic nature of prototheca infection in cats, we think that this question could not be answered easily. Accordingly, we did not put emphasis on these topics in the manuscript, since we did not want to make several hypotheses and speculations. We had added in the discussion a sentence where we mention that since neoplastic diseases are usually immunosuppressive, the cat was likely more prone/susceptible to this systemic localization: “However, the animal had a suspected diagnosis of multifocal lymphoma and therefore it was likely affected by immunological disorders/suppression facilitating the infection and systemic spread of the algal pathogen” (page 7, revised manuscript).
R2.4 Please recheck the manuscript for minor typographical and grammatical errors. Example - line 131 - 0.3 mL/kg.
Reply to R2.4 The manuscript was rechecked and minor typographical and grammatical errors were corrected.
Reviewer 3 Report
Comments and Suggestions for Authors
Thank you for submitting this manuscript and for allowing me the opportunity to read your manuscript to Veterinary Sciences.
I read it very interestingly.
I think that the case in this manuscript is very rare as you stated, but this report is a good value in terms of describing the possibility for the risk to infect to cats in natural environment.
I considered well the scientific contributions of the present manuscript, but the clinical and pathological information is very limited as the author mentioned.
Although the contents are satisfactory, I would like to ask you for some corrections.
1. Please, add the information about cytological features of CSF. Was there any pathogen? only mononuclear cells?
Line 57 “P wickeramii ” should be “P wickerhamii ”.
Author Response
Dear Referee,
herein you can find a point-by-point response to the comments and suggestions.
Thank you for submitting this manuscript and for allowing me the opportunity to read your manuscript to Veterinary Sciences. I read it very interestingly. I think that the case in this manuscript is very rare as you stated, but this report is a good value in terms of describing the possibility for the risk to infect to cats in natural environment. I considered well the scientific contributions of the present manuscript, but the clinical and pathological information is very limited as the author mentioned.
General Reply to R3: We thank the referee for his/her appreciation for the manuscript. The efforts of the co-authors are surely paid back by these considerations. We are aware of the limits of the study and we disclosed them in the text.
R3.1. Although the contents are satisfactory, I would like to ask you for some corrections.
Please, add the information about cytological features of CSF. Was there any pathogen? only mononuclear cells?
Reply to R3.1. A lumbar punctate CFS sample was categorized as positive for pleocytosis in the case of increased protein fraction (>30mg/dL) or increased number of cells (>3 cells/mL) with a predominance of mononuclear cells. In the specific CSF sample, red blood cells were not present. We added this information in the Materials and methods section (page 2, revised manuscript).
Also, we edited a sentence already present in the text: “Examination of the CSF revealed a marked increase in total protein (2432 mg/l, range <300 mg/l), and cytological features consistent with a marked mixed-type pleocytosis (2448 cells/ul, range <3 cells/ul), composed mainly of small and medium-sized lympho-cytes with no red blood cells.” (page 5, revised manuscript)
R3.2. Line 57 “P wickeramii” should be “P wickerhamii”.
Reply to R3.2. This was done.
Reviewer 4 Report
Comments and Suggestions for Authors
Dear all,
I hope this message finds you well.
I suggest minor English corrections and to check my suggestion to change the title of the paper from "Algal infection in a cat with neurological symptoms" to "Prototheca spp. infection in a cat with neurological
Furthermore, I suggest minor English corrections and to check my suggestion to change the title of the paper from "Algal infection in a cat with neurological symptoms" to "Prototheca spp. infection in a cat with neurological symptoms".
Sincerely
Comments on the Quality of English LanguageDear all,
I hope this message finds you well.
I suggest minor English corrections and to check my suggestion to change the title of the paper from "Algal infection in a cat with neurological symptoms" to "Prototheca spp. infection in a cat with neurological symptoms".
Sincerely
Author Response
Dear Referee,
herein you can find a point-by-point response to the comments and suggestions.
I suggest minor English corrections and to check my suggestion to change the title of the paper from "Algal infection in a cat with neurological symptoms" to "Prototheca spp. infection in a cat with neurological signs”
General Reply to R4: We modified the title of the manuscript as “Identification of prototheca from the cerebrospinal fluid of a cat with neurological signs” as suggested. We also checked English throughout the manuscript.
Reviewer 5 Report
Comments and Suggestions for Authors
This manuscript represents an unusual case of feline neurologic disease caused by protothecosis. Certainly is a unfrequent disease in feline, specifically without cutaneous, nasal or ocular presentation.
In addition the study updates Prototheca sp. taxonomy. On the other hand, the cause of the infection has not been confirmed, nor the form of inoculation. Final diagnosis of lymphoma has not been established.
Author Response
Dear Referee,
herein you can find a point-by-point response to the comments and suggestions.
This manuscript represents an unusual case of feline neurologic disease caused by protothecosis. Certainly, is a infrequent disease in feline, specifically without cutaneous, nasal or ocular presentation. In addition, the study updates Prototheca sp. taxonomy. On the other hand, the cause of the infection has not been confirmed, nor the form of inoculation. Final diagnosis of lymphoma has not been established.
General Reply to R5: We thank the referee for his/her appreciation for the manuscript. We highlighted the identification of Prototheca in the CSF of a cat with neurological signs although we could not retrieve the cause of infection.
Round 2
Reviewer 1 Report
Comments and Suggestions for Authors
The redrafting of the manuscript makes it clear that there are limitations that prevent the conclusion that Prototheca is proven to cause this neurological disease. It only evidences the discovery of the infectious agent in CSF and makes it clear that further research is needed.
I agree with this new version of the paper and therefore it can be published as now drafted.